# OpenReview forum: "Does Safety Training of LLMs Generalize to Semantically Related Natural Prompts?"
_NeurIPS.cc/2024/Workshop/SafeGenAi — SafeGenAi Poster_

### Official Review · Reviewer_dFGc · 2024-10-09
**An interesting and effective method for finding adversarial prompts for safety-tuned LLMs, could use more analysis**

**Rating:** 7
**Confidence:** 4

**Review:**

This work proposes to find a set of adversarial prompts i.e. jailbreaking prompts, that are successful on safety tuned LLMs, while remaining close to the set of natural appearing prompts.
The Hypothesis is that safety-tuning in LLMs does not generalize equally to answer and question generation. That is, the authors define a set of unsafe (toxic) answers, and find a set of prompts that elicit such outputs from a safety-tuned LLM.

I found this work to be interesting, and the results of the method are quite good. I found table 3 especially convincing. For the full paper I would be interested in some additional analysis, but I think this should make a good addition to the workshop.

For the full paper I would have the following questions:
- The authors mention: "While we leverage a safety-aligned LLM for question generation, one could alternatively utilize an unaligned LLM for this step. ". I appreciate that the authors want to make the setting more difficult, but in this case I think it would be interesting to contrast the performance / success rate given an unaligned question generator.
- Why are we using Palm2-Otter only for Q->A generation? Performance on other unaligned base models would be interesting.
- Is there any comment one why certain subjects w.r.t. the jailbreak perform better or worse than paraphrasing? Evaluating with an open model where the data mixture is known could yield some insights.
- (comment) I'm mostly missing useful discussion about the phenomenon itself. Why would safety training not generalize A->Q? Why do current defenses fail?

---

### Official Review · Reviewer_nRok · 2024-10-10

**Rating:** 7
**Confidence:** 4

**Review:**

1. **Quality:**
The paper presents an evaluation of the robustness of safety-trained LLMs against natural prompts that are semantically related to toxic prompts. The authors propose a novel approach called Response Guided Question Augmentation (ReG-QA), to systematically assess the generalization of safety fine-tuning in LLMs. The authors have validated the effectiveness of their method through a thorough experimental setup and comparison with existing methods and defenses.


2. **Clarity:**
The paper is well-structured, with clear definitions and explanations of concepts.


3. **Originality:**
The approach to generating semantically related natural prompts using the proposed ReG-QA method is original. This technique provides a new way to probe LLMs beyond typical adversarial attacks or typical prompt-level attacks.


4. **Significance:**
The work is significant as the findings raise important questions about the generalization capabilities of safety alignments in LLMs, suggesting areas for future improvement in model training and evaluation.


I am curious, as shown in Figure 1, whether the ReG-QA method would still be effective if adversarial training were conducted using data generated by methods such as A1 (Adversarial Attacks without semantic constraints, e.g., [1]) or A2 (Adversarial Attacks with semantic constraints, e.g., [2]). This would to some extent verify the out-of-distribution generalization of LLMs after safety training.


[1] Liao, Zeyi, and Huan Sun. "Amplegcg: Learning a universal and transferable generative model of adversarial suffixes for jailbreaking both open and closed llms." arXiv preprint arXiv:2404.07921 (2024).

[2] Paulus, Anselm, et al. "Advprompter: Fast adaptive adversarial prompting for llms." arXiv preprint arXiv:2404.16873 (2024).